# The Neutrophil/Lymphocyte Count Ratio Predicts Mortality in Severe Traumatic Brain Injury Patients

**DOI:** 10.3390/jcm8091453

**Published:** 2019-09-12

**Authors:** Dorota Siwicka-Gieroba, Katarzyna Malodobry, Jowita Biernawska, Chiara Robba, Romuald Bohatyrewicz, Radoslaw Rola, Wojciech Dabrowski

**Affiliations:** 1Department of Anaesthesiology and Intensive Care Medical University of Lublin, 20-954 Lublin, Poland; w.dabrowski5@yahoo.com; 2Faculty of Medicine, University of Rzeszow, 35-959 Rzeszow, Poland; katarzyna.malodobry112@o2.pl; 3Department of Anaesthesiology and Intensive Care Pomeranian University of Szczecin, 71-252 Szczecin, Poland; lisienko@wp.pl (J.B.); romuald.bohatyrewicz@pum.edu.pl (R.B.); 4Department of Anaesthesia and Intensive Care, Policlinico San Martino, IRCCS for Oncology and Neuroscience, 16100 Genova, Italy; kiarobba@gmail.com; 5Department of Neurosurgery with Paediatric Neurosurgery Medical University of Lublin, 20-954 Lublin, Poland; rola.radoslaw@gmail.com

**Keywords:** Neutrophil-lymphocyte count ratio, traumatic brain injury, Extended Glasgow Outcome Score, diffuse axonal injury, cerebral edema

## Abstract

Introduction: Neutrophil-lymphocyte count ratio (NLCR) is a simple and low-cost marker of inflammatory response. NLCR has shown to be a sensitive marker of clinical severity in inflammatory-related tissue injury, and high value of NLCR is associated with poor outcome in traumatic brain injured (TBI) patients. The purpose of this study was to retrospectively analyze NLCR and its association with outcome in a cohort of TBI patients in relation to the type of brain injury. Methods: Adult patients admitted for isolated TBI with Glasgow Coma Score lower than eight were included in the study. NLCR was calculated as the ratio between the absolute neutrophil and lymphocyte count immediately after admission to the hospital, and for six consecutive days after admission to the intensive care unit (ICU). Brain injuries were classified according to neuroradiological findings at the admission computed tomography (CT) as DAI—patients with severe diffuse axonal injury; CE—patients with hemispheric or focal cerebral edema; ICH—patients with intracerebral hemorrhage; S-EH/SAH—patients with subdural and/or epidural hematoma/subarachnoid hemorrhage. Results: NLCR was calculated in 144 patients. Admission NLCR was significantly higher in the non-survivors than in those who survived at 28 days (*p* < 0.05) from admission. Persisting high NLCR value was associated with poor outcome, and admission NLCR higher than 15.63 was a predictor of 28-day mortality. The highest NLCR value at admission was observed in patients with DAI compared with other brain injuries (*p* < 0.001). Concussions: NLCR can be a useful marker for predicting outcome in TBI patients. Further studies are warranted to confirm these results.

## 1. Introduction

Traumatic brain injury (TBI) is a major cause of death and disability worldwide [1]. While primary brain injury is irreversible, secondary damage-consequent to neuronal dysfunction related to trauma induced oxidative stress, ischemia, edema, and inflammatory response is amenable for treatment [2]. Inflammatory response after TBI may be triggered by the presence of injured neuronal cells, and the damage-related inflammatory molecular cascade then proceeds with the release of different proinflammatory cytokines and angiogenic factors with consequent degradation of tight junctions (TJs), cytoskeletal rearrangement, and protein extravasation promoted by adhesion molecules [3]. Additionally, the uncontrolled release of metalloproteinases, inflammatory cytokines, and proteases and the inappropriate activation of endothelial cells may further impair the integrity of the blood-brain barrier (BBB), leading to proteinaceous fluid extravasation into the interstitial space and significant leukocyte infiltration [3,4,5]. The first peak of increased BBB permeability is observed within the first few hours after injury and persists for 3–4 days; a second peak may occur after 5 days as a result of microglial activation [6,7,8]. An experimental study has documented that this process leads to the recruitment of neutrophils to the site of injury in the first hour after injury [9], which may result in white blood cell (WBC) disorders [10].

Peripheral WBC analysis is a simple and low-cost test that provides important information about the general inflammatory response. A peak of WBC has been demonstrated after delayed cerebral ischemia and has been proposed as an independent risk factor for cerebral vasospasm following aneurysmal subarachnoid hemorrhage [11]. Additionally, the neutrophil-to-lymphocyte count ratio (NLCR) has been suggested as a sensitive biomarker to measure the inflammatory status of the immune system in different conditions, such as malignancy [12,13], cardiovascular diseases [14], and stroke [15]. Recent studies have also suggested that NLCR is a useful marker for the prediction of clinical outcome in patients with TBI [16], with high NLCR at admission being associated with poor outcome [17,18]. Furthermore, it is well-recognized that the type and severity of TBI plays a crucial role in the activation of the inflammatory response [19,20]. At present, no data documenting the changes in the NLCR in relation to the type and severity of TBI are available.

The first aim of this retrospective study was to analyze the NLCR and its effect on outcome in relation to the type of brain injury in a cohort of isolated TBI. A secondary aim was to assess the difference in NLCR values between different types of TBI.

## 2. Patients and Methods

This study was approved by the Bioethics Committee of the Medical University in Lublin, Poland, and by the Bioethics Committees of Szczecin, Poland. This retrospective analysis of the NLCR was performed, including consecutive adult patients with isolated severe TBI admitted to the intensive care unit (ICU). Patients aged below 18 years, pregnant women, patients with drug-overdoses, and patients with a history of neoplastic, cardiac, hepatic diseases, or renal diseases were excluded.

### 2.1. ICU Protocol and Treatment

All patients included in the study were sedated with propofol (AstraZeneca, Macclesfield, UK) and fentanyl (Polfa, Warsaw, Poland) and were intubated and mechanically ventilated; 30° head elevation was implemented for all patients. The fraction of inspired oxygen (FiO_2_) was adjusted to maintain oxygen saturation (SpO_2_) between 92% and 98% and regional cerebral oximetry (SrO_2_) higher than 50%. Patients were treated according to the latest Brain Trauma Foundation guidelines [21]. Heart rate (HR), continuous mean arterial pressure (MAP), regional cerebral oximetry (SrO_2_), and SpO_2_ were monitored in all patients. Additionally, hemodynamic variables, such as cardiac output/index (CO/CI), stroke volume variation (SVV), central venous pressure (CVP), systemic vascular resistance index (SVRI), and extravascular lung water index (ELWI) were monitored using EV 1000 platform (Edwards Lifesciences, Irvine, CA, USA). The frontotemporal activity was monitored with electroencephalography (EEG) using a Masimo Root device with a SEDLine monitor (Irvine, CA, USA) or bispectral complete 4-channel monitor (BIS, Medtronic, Minneapolis, MN, USA). Blood potassium, sodium, glucose, and lactate levels were measured 5 times/day, and blood osmolality was measured 1–2 times/day. Continuous norepinephrine infusion and balanced crystalloids (Sterofundin ISO, Melsungen, G) were used to maintain cerebral perfusion pressure (CPP) above 60 mmHg. Patients with intracranial hypertension despite the adoption of basic measures to control and prevent intra-cranial pressure (ICP) received as first instance hyperosmotic therapy with 1.5 g/kg 15% mannitol to reduce ICP. Hyperosmotic therapy was discontinued in patients who had a plasma osmolality higher than 310 mOsm/kg H_2_O. Other medical procedures to control ICP included carbon dioxide (CO_2_) optimization, temperature control, and cerebral spinal fluid withdrawal.

Intracranial space-occupying lesions, i.e., subdural and/or epidural hematomas, were evacuated via craniotomy or craniectomy at neurosurgeon’s discretion. Subsequently, decompressive craniectomy was performed in patients with refractory intracranial hypertension with mass effect and/or cerebral herniation [21].

### 2.2. Study Protocol and Data Collection

Blood samples were routinely collected for full blood count analysis immediately after admission to the hospital and for six consecutive days after admission to the ICU. The NLCR was calculated as the ratio between the absolute neutrophil and lymphocyte count.

Brain injuries were classified according to neuroradiological findings at the admission computed tomography (CT) as DAI—patients with severe diffuse axonal injury; CE—patients with hemispheric or focal cerebral edema; ICH—patients with intracerebral hemorrhage; S-EH/SAH—patients with epidural and/or subdural hematoma/subarachnoid hemorrhage. DAI was diagnosed as the presence of multiple microhemorrhage focal lesions located at the grey-white matter junction and/or in the corpus callosum and/or the brainstem. Additionally, the Glasgow Coma Scale (GCS) at admission and the 8-point the Extended Glasgow Outcome Score (GOSE—Table 1) [22] at 7, 28-days, and 6-months were collected. Patients who withdrew their agreement from participation in this study were excluded from the 6-months outcome analysis. Poor outcome was defined as death or patients with GOSE 2–4.

### 2.3. Statistical Analysis

Data distribution was determined using the Kolmogorov–Smirnov test. Categorical variables were compared using the χ^2^ and Fisher’s exact test, and Yates’ correction was applied. Student’s unpaired t-test was used for normally distributed data analysis, which is presented as the mean and standard deviation (mean ± SD and SEM). Non-parametric data were analyzed using the Wilcoxon single-rank test and the Mann–Whitney U test for initial detection of differences, and these data were presented as a median, quartiles 1 and 3, and IQR—interquartile range. The cut-off points for NLCR were calculated with the use of receiver operator characteristic (ROC) curves with auto-calculated maximum specificity and sensitivity. 

## 3. Results

A total of 144 adult patients aged 18–89 years were included in the present study. The main demographic data are presented in Table 2. The median admission GCS score was five (IQR: 3–6). In the DAI group, 7 and 28-day mortality were 10.3% and 34.5%, respectively. After 6 months, six patients were excluded from the study because they or their families did not agree to respond to the phone interview (Table 2). 

The median admission NLCR was 11.74 (IQR: 6.79–19.23) in the studied population. The highest NLCR value at admission was detected in the DAI group, compared to the CE, ICH, and S-EH/SAH groups (Figure 1). Admission NLCR was statistically higher in the non-survivors than in those who survived at 28 days from admission (Figure 2).

Among the patients with CE, the 28-day mortality was 17.65%. Eight patients withdrew their agreement to this study after discharge from ICU or at the phone interview. In the ICH group, 28-day mortality was 52.63%, and only nine patients responded at the 6 months phone interview. In the SAH or subdural and/or epidural hematoma group, 7 and 28-day mortality were 16.13% and 4.84%, respectively. After 6 months, only 28 patients responded to the phone interview. 

ICH patients presented the highest 6-month mortality, compared to the other groups: S-EH/SAH (χ^2^ = 6.94, *p* < 0.01, χ^2^ with Yates’ correction = 5.49 for *p* < 0.05), and CE (χ^2^ = 7.08, *p* < 0.01, χ^2^ with Yates’ correction = 5.52 for *p* < 0.05). The mortality rate was higher in the DAI group compared to the S-EH/SAH group (χ^2^ = 5.29, *p* < 0.05, χ^2^ with Yates’ correction = 4.21 for *p* < 0.05) and CE group (χ^2^ = 5.49, *p* < 0.05, χ^2^ with Yates’ correction = 4.27 for *p* < 0.05). Additionally, the mortality rate was higher in the CE group than in the S-EH/SAH group (χ^2^ = 7.17, *p* < 0.01, χ^2^ with Yates’ correction = 5.7 for *p* < 0.05). GOSE was significantly different between ICH group and S-EH/SAH, DAI, and S-EH/SAH groups (*p* < 0.05).

Two patients were dead at 28 days after TBI and discharge from ICU (GOSE 1) (Table 3). NLCR was significantly higher in patients with GOSE 1, GOSE 2, and GOSE 3 compared to the other groups during the studied period (Table 3). NLCR of 15.63 was calculated as the cut-off point for a significant increase in the 28-day mortality risk (Figure 3).

## 4. Discussion

The present study demonstrated the role of NLCR as a predictor of poor outcome in patients with TBI. NLCR higher than 15.63 at admission was found to be a predictor for 28-day mortality. Additionally, significantly higher NLCR value during the first week of treatment showed to be correlated with severe disability in patients with TBI. Admission NLCR was significantly higher in the DAI group compared to the CE, ICH, and S-EH/SAH groups.

Neutrophils constitute the 50–70% of the total circulating leukocytes, and their recruitment and activation is a characteristic for an early inflammatory response following TBI [23]. Neutrophils infiltration into the injured sub-endothelial space plays an important role in the increase of BBB permeability. 

Disrupted vascular wall induces increased plasma and molecules leakage into the extravascular space, which intensifies cerebral edema [3,5,24]. Additionally, activated matrix metalloproteinases (MMPs), such as MMP-2, MMP-3, MMP-7, and MMP-9, induce BBB disruption in experimental ischemic stroke and severe inflammatory response following septic shock [25,26,27]. Yang et al. documented that early disruption of the BBB caused an increased expression of mRNA for MT1-MMP and furin, which activated MMP-2; finally, this process led to tight junctions (TJs) injury [25]. On the other hand, MMP-9 activity increases after 24–72 hours after BBB injury event disrupting TJs and basal lamina proteins [25,27]. Notably, a major enzyme of neutrophil azurophilic granules is myeloperoxidase MMP-9-positive, thus suggesting that neutrophils play a crucial role in BBB disruption [27,28]. A recent clinical study using cerebral microdialysis documented that the expression of neutrophil collagenase was associated with increased intracranial pressure and was significantly higher in non-survivors than in survivors after TBI [29]. Additionally, increased expression of MMP-9 in the pericontusional area of brain injury within 72 hours was associated with increased risk of refractory vasogenic edema and pericontusional hemorrhagic progression [29]. Other authors suggested that prolonged neuroinflammation impairs regeneration and promotes secondary injury and neurodegeneration, leading to poor outcome [30]. Also, a rapid elevated blood neutrophils levels may result from a TBI-induced increase of catecholamines and glucocorticosteroids [10,17,31,32,33].

In the present study, NLCR at admission was higher in non-survivors than in survivors. Also, in patients with poor outcome, NLCR in the first 6 days after admission was higher than in patients with a good outcome. Based on our findings, we can speculate that higher NLCR might be related to persistent inflammation in the pericontusional brain; however, this hypothesis has to be confirmed in further studies.

Previous studies have explored the role of NLCR as prognostic factors after brain injury [10,15,16,17]. Chen et al. demonstrated a higher NLCR in TBI patients with unfavorable outcome compared to NLCR in patients with favorable outcome, with a cut-off point identified as 18.16 [10]. AL-Mufti et al. found that admission NLCR higher than 5.9 predicted a twofold increased risk of delayed cerebral ischemia after aneurysmal subarachnoid hemorrhage [18]. Additionally, persistently high NLCR was associated with a lower 6 months GOS after TBI [17]. Our findings are in agreement with those observations. Higher NLCR was observed in patients who died after 28 days from admission and in those with unfavorable neurological outcome after 3 months. In contrast, some authors did not confirm the sensitivity of NLCR in predicting poor outcome in TBI patients, suggesting that other factors, such as the international normalized ratio (INR), have a higher sensitivity as prognostic markers when compared to NLCR [33,34,35]. 

In the present study, the highest NLCR was observed in patients with DAI. Indeed, TBI-related DAI is associated with a massive neuroinflammatory response with neutrophils infiltration to the cortex, subcortical white matter, and areas of axonal injury, leading to complex cognitive disorders [19,36,37]. An increase in the TBI-related inflammatory response begins 2–3 hours after injury and reaches maximal levels in 12–24 hours [37,38,39,40]. Notably, the severity of TBI-related neuroinflammation also depends on microglial activation. Proinflammatory mediators promote the M1 type of microglia, which intensifies the release of proinflammatory cytokines and production of reactive oxygen species [41]. Microglia-released cytokines prolong astrocyte activation, leading to the formation of glial scars. These glial scars limit axonal regeneration and prevent new functional connections between neuronal networks, resulting in cognitive deficits in memory, executive function, and attention [42]. These mechanisms may also explain the high incidence of poor neurological outcome in patients with severe DAI; indeed, a clinical study demonstrated that the number of brain lesions identified on MRI was associated with an increased risk of mortality [43].

### Limitations

This study has several limitations. One important limitation of the study is the small number of respondents who agreed to respond to our phone interview (only nine and 12 patients or their family responded to the questionnaire in ICH and DAI groups, respectively). This fact significantly limited the power of our statistical analysis. Second, in the present study, we aimed to demonstrate the usefulness of a simple marker-NLCR-as a prognostic factor of outcome after TBI; however, the inclusion of other markers, and in particular of the brain-damage biomarkers or inflammatory cytokines, could have added further information and strengthened our results.

The high usefulness of NLCR and biomarkers has been also previously described as predictors of delirium in patients with acute ischemic stroke [44]. Indeed, brain injury following trauma or peri-traumatic ischemia increases the concentration of specific biomarkers, such as S100β protein, neuronal-specific enolase (NSE), ubiquitin C-terminal hydrolase-L1 (UCH-L1), and glial fibrillary acidic protein (GFAP) [45,46,47,48]. Some of those biomarkers correlated with Glasgow Outcome Score at 6-month and with mortality in TBI patients [46]. Some authors have also shown that inflammatory cytokines are sensitive in the prognosis of post-TBI recovery and significantly correlated with 6-month outcome [47,49]. Noteworthy, the severity of TBI is commonly assessed by CT examinations, which cannot provide any information about the metabolic and inflammatory response after brain injury. The severity of neuroinflammatory-related trauma can be only provided through advanced imaging or biomarkers; however, biomarkers sample and analysis are frequently expensive [50,51]. On the contrary, NLCR is a low-cost and very simple marker, which can be determined in every hospital. 

## 5. Conclusions

In our study, we demonstrated the potential usefulness of NLCR for the definition of prognosis of brain injuries. The neutrophil-to-lymphocyte count ratio can also be considered as a sensitive marker for predicting outcome according to the type of TBI; however, further studies should be performed to establish the best prognostic marker for patients with TBI.

## Figures and Tables

**Figure 1 jcm-08-01453-f001:**
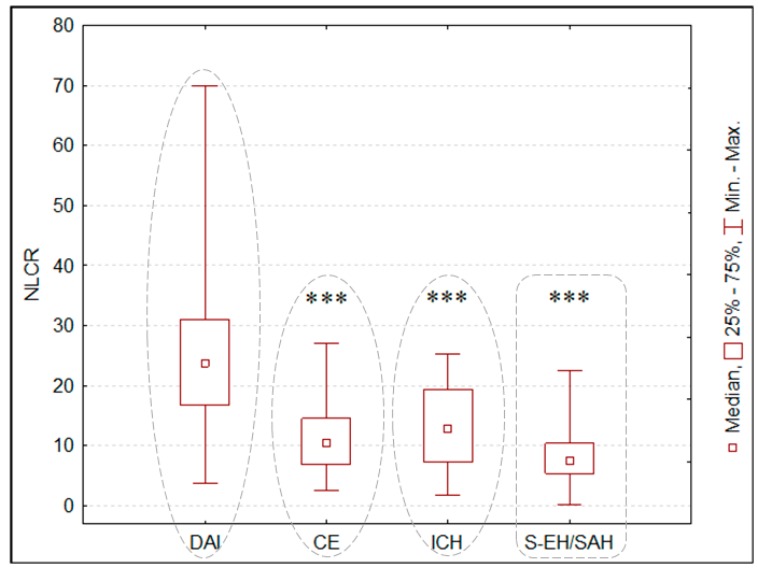
Neutrophil-to-lymphocyte count ratio (NLCR) in relation to different types of traumatic brain injury. *** *p* < 0.001 compared with baseline. DAI—diffuse axonal injury, CE—cerebral edema, ICH—intracerebral hemorrhage, S-EH/SAH—subarachnoid hemorrhage, epidural and/or subdural hematoma.

**Figure 2 jcm-08-01453-f002:**
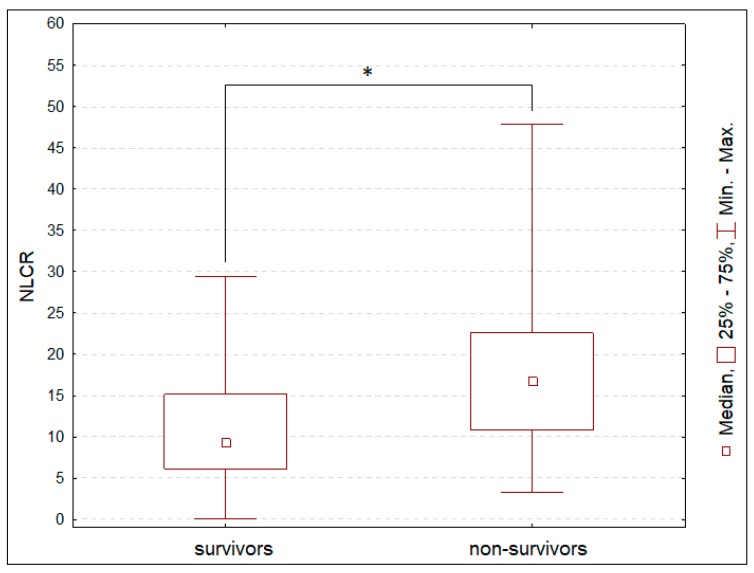
Differences in neutrophil-to-lymphocyte count ratio (NLCR) in patients, who survived at 28 days, and those, who did not survive. * *p* < 0.05.

**Figure 3 jcm-08-01453-f003:**
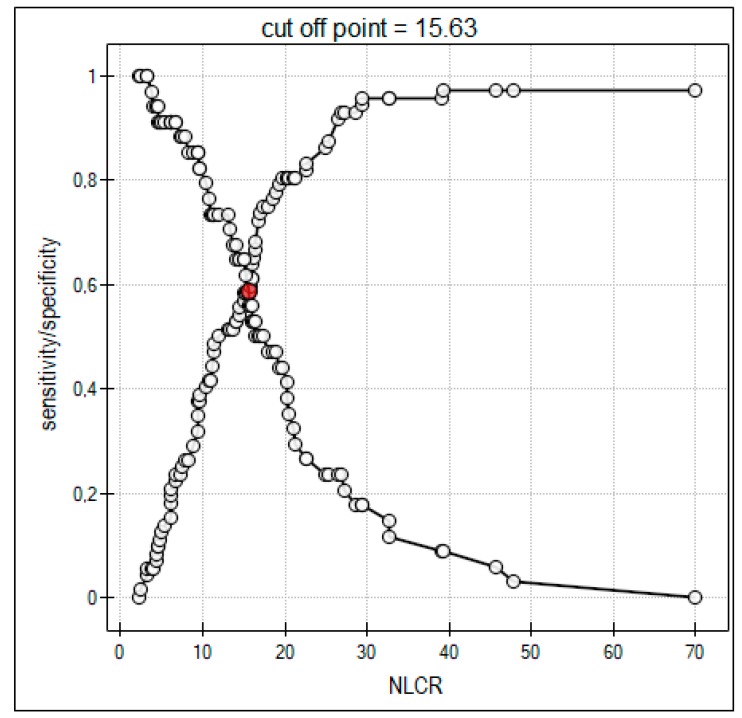
The ROC (receiver operator characteristic) curve for the neutrophil-to-lymphocyte count ratio (NLCR) in accordance with the mortality risk. An increase in NLCR with a cut-off point of 15.63 increases risk of death.

**Table 1 jcm-08-01453-t001:** The Extended Glasgow Outcome Score measured 6 months after TBI (traumatic brain injury).

Points	Clinical Condition
1	Death
2	Vegetative state (VS)
3	Lower severe disability (SD-)
4	Upper severe disability (SD+)
5	Lower moderate disability (MD-)
6	Upper moderate disability MD+)
7	Lower good recovery (GR-)
8	Upper good recovery (GR+)

**Table 2 jcm-08-01453-t002:** Demographic data.

TBI	Age	Sex	GCS	28-Day Mortality	GOSE 6-Months Outcome	RQ	Mean GOSE
Total (*n* = 144)	48 (IQR: 32–59)	26 female 118 male	5 (IQR: 3–6)	42 patients (29.17%)	Death	2	5.1 ± 2.1
					VS	8	
					SD-	10	
					SD+	7	
					MD-	11	
					MD+	7	
					GR-	13	
					GR+	10	
DAI (*n* = 29)	50 (IQR: 35–57)	4 female 25 male	4.6 ± 1.1	13 patients (44.83%)	Death	2	5.4 ± 1.98
					VS	2	
					SD-	0	
					SD+	0	
					MD-	2	
					MD+	2	
					GR-	4	
					GR+	0	
CE (*n* = 34)	44 (IQR: 32–57)	7 female 27 male	5.3 ± 1.96	6 patients (17.65%)	Death	0	4.9 ± 1.79
					VS	2	
					SD-	4	
					SD+	0	
					MD-	7	
					MD+	2	
					GR-	4	
					GR+	1	
ICH (*n* = 19)	54 (IQR: 43–59)	2 female 17 male	4.3 ± 1.5	10 patients (52.63%)	Death	0	3.7 ± 1.3
					VS	2	
					SD-	3	
					SD+	2	
					MD-	1	
					MD+	1	
					GR-	0	
					GR+	0	
S-EH/SAH (*n* = 62)	44 (IQR: 29–60)	10 female 52 male	5.3 ± 1.8	13 patients (20.97%)	Death	0	5.8 ± 2.14
					VS	2	
					SD-	3	
					SD+	6	
					MD-	1	
					MD+	2	
					GR-	5	
					GR+	9	

RQ—respondents to questionnaire, DAI—diffuse axonal injury, CE—cerebral edema, ICH—intracerebral hemorrhage, S-EH/SAH—subarachnoid hemorrhage, epidural and/or subdural hematoma, GCS—Glasgow Coma Scale, GOSE—Extended Glasgow Outcome Score.

**Table 3 jcm-08-01453-t003:** Changes in the mean value of neutrophil-to-lymphocyte count ratio (NLCR) during the first 6 days of treatment in accordance with 6-months Extended Glasgow Outcome Score (GOSE). * *p* < 0.05, ** *p* < 0.01—comparison with admission day (Day 0).

GOSE Scores	Day 0	Day 1	Day 2	Day 3	Day 4	Day 5	Day 6
GOSE 1 and GOSE 2	17.79 ± 5.7 (SEM = 1.8)	15.92 ± 3.97 (SEM = 1.3)	12.05 * ± 4.2 (SEM = 1.3)	12.69 ± 7.5 (SEM = 2.4)	11.93 ± 5.2 (SEM = 1.6)	13.02 ± 4.8 (SEM = 1.5)	10.11 ** ± 3.1 (SEM = 0.98)
GOSE 3	13.66 ± 6.3 (SEM = 2.0)	9.23 ± 4.2 (SEM = 1.3)	9.23 ± 3.9 (SEM = 1.2)	11.2 ± 6 (SEM = 1.9)	10.06 ± 4.23 (SEM = 1.3)	9.95 ± 3.2 (SEM = 1.0)	8.59 ± 3.7 (SEM = 1.2)
GOSE 4	9.55 ± 5.6 (SEM = 2.0)	5.96 ± 2.4 (SEM = 0.84)	5.23 * ± 3.6 (SEM = 1.3)	5.59 ± 2.4 (SEM = 0.85)	7.7 ± 4.6 (SEM = 1.6)	6.01 ± 3.3 (SEM = 1.2)	5.94 ± 3.2 (SEM = 1.1)
GOSE 5	14.4 ± 10.8 (SEM = 3.0)	7.12 * ± 2.8 (SEM = 0.8)	5.54 * ± 2.1 (SEM = 0.6)	6.03 * ± 2.2 (SEM = 0.6)	7.51 * ± 1.8 (SEM = 0.5)	9.52 ± 6.3 (SEM = 1.7)	7.08 * ± 2.9 (SEM = 0.8)
GOSE 6	12.77 ± 8.7 (SEM = 3.3)	16.29 ± 19.3 (SEM = 7.2)	11.5 ± 10.6 (SEM = 4.0)	6.7 ± 4.97 (SEM = 1.9)	8.23 ± 3.7 (SEM = 1.4)	5.72 * ± 2.1 (SEM = 0.8)	5.99 * ± 2.1 (SEM = 0.8)
GOSE 7	17.9 ± 16.9 (SEM = 4.7)	7.69 ** ± 3.9 (SEM = 1.1)	5.47 ** ± 2.1 (SEM = 0.6)	5.61 ** ± 2.6 (SEM = 0.7)	5.48 ** ± 2 (SEM = 0.6)	4.5 ** ± 1.3 (SEM = 0.4)	5.12 ** ± 2.1 (SEM = 0.6)
GOSE 8	15.53 ± 18.8 (SEM = 5.7)	7.57 ** ± 4.9 (SEM = 1.5)	6.47 * ± 3.6 (SEM = 1.1)	4.98 ** ± 2.8 (SEM = 0.9)	5.43 * ± 3.5 (SEM = 1.1)	4.01 ** ± 1.5 (SEM = 0.5)	4.94 * ± 2.1 (SEM = 0.6)

NLCR in GOSE 1 and GOSE 2 groups was significantly higher than NLCR in GOSE 3 group only at day 1 (*p* < 0.05). Additionally: NLCR in patients with GOSE 1 and GOSE 2 was significantly higher than NLCR in patients with: GOSE 4 at days 0 (*p* < 0.01), 1 (*p* < 0.001), 2 (*p* < 0.01), 3 (*p* < 0.001), 5 (*p* < 0.01), and 6 (*p* < 0.05); GOSE 5 from days 2 to 5 (*p* < 0.001) and at day 6 (*p* < 0.05); GOSE 6 at days 3 (*p* < 0.05), 5 (*p* < 0.001), and 6 (*p* < 0.05); GOSE 7 from day 1 to 6 (*p* < 0.001), and GOSE 8 at days 0 (*p* < 0.05), 1 (*p* < 0.01), 2 (*p* < 0.05), 3 (*p* < 0.01), 4 (*p* < 0.001), and 5 and 6 (*p* < 0.001). Similar differences were noted in NLCR in patients with GOSE 3 and the other patients. There were no significant differences in NLCR between patients with GOSE 4, GOSE 5, GOSE 6, GOSE 7, and GOSE 8 in all studied days.

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
