# Peer review of "The Neutrophil/Lymphocyte Count Ratio Predicts Mortality in Severe Traumatic Brain Injury Patients"

_jcm, 2019, doi:10.3390/jcm8091453_

Round 1
Reviewer 1 Report
we read with great interest the study by Dabrowski group assessing the levels of the NLCR as aprognostic value in TBI, where they evaluated its effect on outcome in relation to the type of brain injury.
the hypothesis of the study is that the NLCR differs between different types of TBI and they had different groups of brain insults included.
there are few comments,
the study requires a major positive control that can compare the NLRC to, such as using a known TBI biomarker like UCH-L1 or GFAp etc..
the study would benefit in any measurement of cytokine is included as it will reflect on the mecahnistic value of the NLRC.
A full demographic of patients is needed
figure 1 is hard to understand and need to be more elaborated
Author Response
Dear Reviewers
We thank the reviewers for their further review of our manuscript titled: "The neutrophil/lymphocyte count ratio predicts mortality in severe traumatic brain injury patients". We have modified the manuscript per the reviewers' comments, marking all changes and corrections as red text. We have provided an English review thorough the manuscript. Our point by point responses are also shown below.
Comment of Reviewer 1
We read with great interest the study by Dabroski group assessing the levels of the NLCR as a prognostic value in TBI, where they evaluated its effect on outcome in relation to the typy of brain injury. The hypothesis of the study is that the NLCR differs between different types of TBI and they had different groups of brain insult included. There are few comments,
The study requires a major positive control that can compare the NLCR to, such as using a known TBI biomarker like UCH-L1 or GFAp etc.
Answer: Thank You for your this comment. Indeed, it would have confirmed much more stronger a credibility of NLCR compared it with S100beta protein, Tau protein, UCH-L1 or GFAP. Unfortunately, we were not able to assess those biomarkers in our study. However, we would like to mark a high heterogeneity of brain injury, which have a huge effect on all biomarkers. Additionally, some of them have been also found in extracranial tissue (S100beta has been found in mediastinum, NSE – in red blood cells etc), which may limit their credibility in injured patients. Nevertheless, we fully agree with your comment and we have marked this fact in the limitation section. We will continue our study and we will compare NLCR with some of popular, specific TBI biomarkers.
The study would benefit in any measurement of cytokine is included as it will reflect on the mechanistic value of the NLCR.
Answer: Thank you for this comment. We agree with your suggestion. We believed, that a persistent neuroinflammation (with elevated cytokine concentration) plays a crucial role for poor outcome. We marked this fact in the limitations. Your suggestion will be provided for continued study.
A full demographic of patients is needed.
Answer: We add this data in the first row in table 1.
Figure 1 is hard to understand and need to be more elaborated.
Answer: Thank you for this comment. We add a few sentences to figure’s legend for a better understanding.
Reviewer 2 Report
Congratulations on investigating NLCR as part of a well designed study. This may be excellent for a resource poor situation. The data is consistent with role of persistent inflammation and TBI outcomes. Given that new biomarkers for TBI have been approved by FDA (https://www.accessdata.fda.gov/cdrh_docs/reviews/DEN170045.pdf), it would be reasonable to continue your investigation's in parallel to a biomarker analysis. Such a parallel could help send TBI patients with high NLCR for further imaging or enrollment in appropriate trial even if the CT was negative.
The authors present an interesting low-cost marker of inflammatory response to TBI, that also happens to be routinely done for the first week during hospitalization. Examining the whole blood and assessing the neutrophil-lymphocyte count ratio (NLCR) serves as a sensitive marker of severity in inflammatory-related tissue injury with certain values associated with poor outcome. This is a retrospective analysis of NLCR, perhaps will lead to a prospective study in real world to improve triage if automated. With an appropriate sample size of 144 patients. The authors show that higher NLCR values on admission day were in patients with DAI than those with CE, ICH and S-EH/SAH (p < 0.001). High NLCR was statistically higher in the non-survivors than in those who survived at least 28 days (p < 0.05), admission day NLCR higher than 15.63 predicted the death within 28 days. Persisted higher NLCR was associated with poor outcome. Authors conclude that NLCR can be useful for diagnosing the severity of the brain injuries. The NLCR can be considered a marker for predicting outcome in TBI patients, perhaps triage and invest in patients that may survive 28 days post injury.
Table 2 Show distribution of 28 patients at 6 month follow up with NLCR.
Figure 1: Please label legends open and filled circles correspond to which type of TBI.
Figure 2: Show a scatter plot box-whisker rather than Median to give readers idea about the population distribution.
Table 3: Instead of Table 3 just data, analyze by 2 way ANOVA and pinpoint NLCR-day and outcome relationship.
Figure 3: Is same as Fig 1, not an ROC curve as in legend, please fix
Compare means ± SEM between groups rather than SD. SD provides variation information not population means
If ICH had high mortality at 6 months, what was their NCLR predictive of death, this undermines the conclusion that NCLR is predictive of death on day 28.
Please use other feedback on use of FDA approved biomarkers as actual association with TBI outcome
Author Response
Dear Reviewers
We thank the reviewers for their further review of our manuscript titled: "The neutrophil/lymphocyte count ratio predicts mortality in severe traumatic brain injury patients". We have modified the manuscript per the reviewers' comments, marking all changes and corrections as red text. We have provided an English review thorough the manuscript. Our point by point responses are also shown below.
Comment of Reviewer 2
Congratulations on investigating NLCR as part of well designed study. This may be excellent for a resource poor situation. The data is consistent with role of persistent inflammation and TBI outcomes. Given that new biomarkers for TBI have been approved by FDA (https://www.accessdata.fda.gov/cdrh_docs/reviews/DEN170045.pdf) it would be reasonable to continue your investigation's in parallel to a biomarker analysis. Such a parallel could help send TBI patients with high NLCR for further imaging or enrollment in appropriate trial even if the CT was negative. The authors present an interesting low-cost marker of inflammatory response to TBI, that also happens to be routinely done for the first week during hospitalization. Examining the whole blood and assessing the neutrophil-lymphocyte count ratio (NLCR) serves as a sensitive marker of severity in inflammatory-related tissue injury with certain values associated with poor outcome. This is a retrospective analysis of NLCR, perhaps will lead to a prospective study in real world to improve triage if automated. With an appropriate sample size of 144 patients. The authors show that higher NLCR values on admission day were in patients with DAI than those with CE, ICH and S-EH/SAH (p < 0.001). High NLCR was statistically higher in the non-survivors than in those who survived at least 28 days (p < 0.05), admission day NLCR higher than 15.63 predicted the death within 28 days. Persisted higher NLCR was associated with poor outcome. Authors conclude that NLCR can be useful for diagnosing the severity of the brain injuries. The NLCR can be considered a marker for predicting outcome in TBI patients, perhaps triage and invest in patients that may survive 28 days post injury.
New comments of reviewer 2
The authors present an interesting low-cost marker of inflammatory response to TBI, that also happens to be routinely done for the first week during hospitalization. Examining the whole blood and assessing the neutrophil/lymphocyte count ratio (NLCR) serves as a sensitive marker of severity in inflammatory-related tissue injury with certain values associated with poor outcome. This is a retrospective analysis of NLCR, perhaps will lead to a prospective study in real world to improve triage if automated. With an appropriate sample size of 144 patients. The authors show that higher NLCR values on admission day were in patients with DAI than those with CE, ICH and S-EH/SAH (p < 0.001). High NLCR was statistically higher in the nonsurvivors than in those who survived at least 28 days (p < 0.05), admission day NLCR higher than 15.63 predicted the death within 28 days. Persisted higher NLCR was associated with poor outcome. Authors conclude that NLCR can be useful for diagnosing the severity of the brain injuries. The NLCR can be considered a marker for predicting outcome in TBI patients, perhaps triage and invest in patients that may survive 28 days post injury. Table 2 Show distribution of 28 patients at 6 month follow up with NLCR
Figure 1: Please label legends open and filled circles correspond to which type of TBI.
Answer: Thank You for this comment. We added a circle to figure 1 and added a few sentences to the legend of figure 1.
Figure 2: Show a scatter plot box-whisker rather than Median to give readers idea about the population distribution.
Answer: Thank you for this comment. We fully changed figure 2.
Table 3. Instead of Table 3 just data, analyze by 2 way ANOVA and pinpoint NLCR-day and outcome relationship. Thank You for this suggestion. We performed it but we did not include it in the text as the samples size is too small, and this has been included in the limitations.
Figure 3: Is same as Fig 1, not an ROC curve as in legend, please fix.
Answer: Thank you. It was our mistake to upload another figure to manuscript during the first submission. This was corrected.
Compare means ± SEM between groups rather than SD. SD provides variation information not population means. 2
Answer: Thank You for this suggestion. It was done and illustrated in table 3.
If ICH had high mortality at 6 months, what was their NCLR predictive of death, this undermines the conclusion that NCLR is predictive of death on day 28.
Answer: Thank you for this comment. Indeed, the 6 month mortality was higher in ICH group, however we did not know a reason of this mortality. A lot of these patients were discharged from our hospital. We assessed such mortality rate based only on phone interview.
Please use other feedback on use of FDA approved biomarkers as actual association with TBI outcome.
Answer: Thank You for this comment. We marked this fact in the limitation section.